# Secondary Malignancy Risk Following Proton vs. X-ray Radiotherapy of Thymic Epithelial Tumors: A Comparative Modeling Study of Thoracic Organ-Specific Cancer Risk

**DOI:** 10.3390/cancers14102409

**Published:** 2022-05-13

**Authors:** Laila König, Juliane Hörner-Rieber, Matthew Forsthoefel, Peter Haering, Eva Meixner, Tanja Eichkorn, Anna Krämer, Thomas Mielke, Eric Tonndorf-Martini, Matthias F. Haefner, Jürgen Debus, Jonathan W. Lischalk

**Affiliations:** 1Department of Radiation Oncology, Heidelberg University Hospital, 69120 Heidelberg, Germany; juliane.hoerner-rieber@med.uni-heidelberg.de (J.H.-R.); eva.meixner@med.uni-heidelberg.de (E.M.); tanja.eichkorn@med.uni-heidelberg.de (T.E.); anna.kraemer@med.uni-heidelberg.de (A.K.); thomas.mielke@med.uni-heidelberg.de (T.M.); eric.tonndorf-martini@med.uni-heidelberg.de (E.T.-M.); matthias.haefner@strahlentherapie-speyer.de (M.F.H.); juergen.debus@med.uni-heidelberg.de (J.D.); 2Heidelberg Institute of Radiation Oncology (HIRO), 69120 Heidelberg, Germany; 3National Center for Tumor Diseases (NCT), 69120 Heidelberg, Germany; 4Heidelberg Ion-Beam Therapy Center (HIT), Department of Radiation Oncology, Heidelberg University Hospital, 69120 Heidelberg, Germany; 5Clinical Cooperation Unit Radiation Oncology, German Cancer Research Center (DKFZ), 69120 Heidelberg, Germany; 6Department of Radiation Oncology, Radiotherapy Centers of Kentuckiana, Louisville, KY 47130, USA; matthew.forsthoefel@usoncology.com; 7Department of Radiation Medicine, MedStar Georgetown University Hospital, 3800 Reservoir Road NW, Washington, DC 20007, USA; jonathan.lischalk@nyulangone.org; 8Department for Medical Physics in Radiation Oncology, German Cancer Research Center (DKFZ), 69120 Heidelberg, Germany; p.haering@dkfz-heidelberg.de; 9German Cancer Consortium (DKTK), Partner Site Heidelberg, 69120 Heidelberg, Germany; 10Department of Radiation Oncology, Perlmutter Cancer Center, New York University Langone Hospital—Long Island, 150 Amsterdam Ave., New York, NY 10023, USA

**Keywords:** thymoma, thymic carcinoma, radiation-induced cancers, proton therapy, photon radiotherapy, intensity-modulated radiotherapy (IMRT)

## Abstract

**Simple Summary:**

Proton beam radiotherapy (PBT) offers the possibility to significantly reduce dose to surrounding organs at risk due to their physical advantages compared to X-ray based techniques. The aim of this analysis was to demonstrate whether PBT reduces secondary malignancy risks in patients with thymic malignancies compared to 3D conformal and intensity-modulated radiotherapy with photons. By using two different mechanistic calculation models we could demonstrate significant reductions of secondary malignancy risks with the use of PBT for all independent thoracic organs analyzed with the exception of the thyroid gland. This technology-driven improvement might translate into clinically relevant benefits for patients with thymic malignancies.

**Abstract:**

Background: Proton beam radiotherapy (PBT) offers physical dose advantages that might reduce the risk for secondary malignancies (SM). The aim of the current study is to calculate the risk for SM after X-ray-based 3D conformal (3DCRT) radiotherapy, intensity-modulated radiotherapy (IMRT), and active pencil beam scanned proton therapy (PBS) in patients treated for thymic malignancies. Methods: Comparative treatment plans for each of the different treatment modalities were generated for 17 patients. The risk for radiation-induced SM was estimated using two distinct prediction models—the Dasu and the Schneider model. Results: The total and fatal SM risks estimated using the Dasu model demonstrated significant reductions with the use of PBS relative to both 3DCRT and IMRT for all independent thoracic organs analyzed with the exception of the thyroid gland (*p* ≤ 0.001). SM rates per 10,000 patients per year per Gy evaluated using the Schneider model also resulted in significant reductions with the use of PBS relative to 3DCRT and IMRT for the lungs, breasts, and esophagus (*p* ≤ 0.001). Conclusions: PBS achieved superior sparing of relevant OARs compared to 3DCRT and IMRT, leading to a lower risk for radiation-induced SM. PBS should therefore be considered in patients diagnosed with thymic malignancies, particularly young female patients.

## 1. Introduction

Thymic malignancies are rare tumors of the anterior mediastinum, which are represented by a variety of histopathologic classifications but broadly fall into two groups—thymoma and thymic carcinoma [1]. In patients diagnosed with thymic malignancies, extended survivorship is very common given the relatively young age at diagnosis, relative minimal medical comorbidities, and the long natural history of thymic tumors. One such feared long-term complication of antineoplastic treatment is the development of a secondary malignancy. Interestingly, there seems to exist an increased risk of secondary malignancies (SM) in patients with thymoma in general, which may be related to the combination of immune dysregulation and environmental or genetic factors [2,3,4]. In fact, Hamaji et al. report data demonstrating patients with completely resected thymoma are at a higher risk of SM versus a recurrence of originally diagnosed thymoma [5].

The backbone of treatment for thymic malignancies is surgical resection. However, adjuvant radiotherapy is often recommended and is dictated primarily by the completeness of surgery and the differentiation between thymoma and thymic carcinoma [6]. In detail, current guidelines recommend adjuvant radiotherapy in the case of incomplete resection and capsular invasion for higher Masaoka Koga stages [6] and NCCN guidelines (Thymomas and Thymic carcinomas, Version 1.2022, https://www.nccn.org/professionals/physician_gls/pdf/thymic.pdf (accessed on 21 April 2022)). Due to the rarity of the disease, prospective randomized trials are limited, and thus, robust level one evidence is oftentimes lacking to assist in management decisions. To this end, adjuvant radiotherapy has not been demonstrated in a randomized fashion to improve upon overall survival [7]. One possible reason for the lack of overall survival improvement with radiotherapy may be the toxicity profile caused by the utilization of more antiquated radiation techniques such as 3DCRT [8]. Modern data utilizing intensity-modulated radiation therapy (IMRT) have demonstrated an ability to deliver high-dose radiotherapy to target volumes while mitigating damage to the surrounding normal structures and has been associated with improvements in overall survival in this patient population [9]. Proton beam therapy (PBT) may widen the therapeutic window even further relative to advanced X-ray-based techniques in this disease site. Therefore, the use of protons is constantly increasing. Protons have advantageous physical properties with an inverse depth-dose profile. Only a very low dose is absorbed by the healthy tissue in the entrance channel and the maximum dose is deposited in the so-called Bragg-peak with a very steep dose fall off behind the tumor volume. The geometric location within the anterior mediastinum allows anterior oblique proton beams to be delivered to the target volumes while, in some cases, completely sparing posterior structures from radiation exposure including the heart, lungs, and esophagus as well as lateral structures such as the breast tissue in female patients [10,11,12].

Data on PBT for thymic malignancies is scarce. Mercado et al. report their experience with a small cohort of 22 patients and only early clinical outcomes with a median follow up of only 13 months [8]. As a consequence, radiobiological modeling is necessary to estimate the rates of radiation-induced cancers in an effort to improve upon management decisions from a radiotherapy standpoint. In this manuscript we use novel radiobiological models to determine the risk of SM based on dosimetric data for patients diagnosed with thymic malignancies using three different treatment modalities including 3D conformal radiotherapy (3DCRT), intensity-modulated radiotherapy (IMRT) calculated as volumetric modulated arc therapy (VMAT), and active pencil beam scanned proton therapy (PBS) from two radiotherapy centers.

## 2. Materials and Methods

### 2.1. Patient Selection and Treatment Planning

Patients who were diagnosed with histologically confirmed thymic malignancies were included in this study. Patients were treated at one of two large centers: (1) Heidelberg University Hospital, Germany, and (2) Georgetown University Hospital, Washington, DC, United States. This study was approved by each of the institutional ethical review committees (IRB No. S-004/2017 (Heidelberg) and 2017-0695 (Washington)). Target delineation and treatment planning has been previously described in detail [13,14]. In short, all patients underwent a 4D-planning computed tomography (CT) with a slice thickness of 3 mm in the supine position for qualitative analysis of impacts of the respiratory motion. Delineation of target volumes was performed in accordance with current guidelines [10]. The gross tumor volume (GTV) included the macroscopic disease present on the planning CT and on all further available imaging techniques (e.g., magnetic resonance imaging (MRI), Positron emission tomography (PET) CT-scans). The GTV was enlarged using a 5–10 mm expansion to form the clinical target volume (CTV) respecting anatomic boundaries. In cases of completely resected tumors, the CTV comprised the preoperative tumor extension and the area of the surgical dissection according to the surgical report and/or review by the treating cardiothoracic surgeon. If a relevant movement was noticed, an additional ITV was contoured to take this movement into consideration. The planning target volume (PTV) was generated by applying a 3–5 mm isotropic expansion of the CTV. On all datasets, organs at risk (OARs) including the heart, bilateral (total) lungs, esophagus, thyroid gland, spinal cord, and breasts (in female patients) were contoured.

Treatment planning of photon plans was performed using either Oncentra External Beam, version 4.5 (Elekta, Stockholm, Sweden) or RayStation version 8A using Monte Carlo planning (RaySearch Laboratories, Stockholm, Sweden). Proton therapy planning was performed with either Syngo RT Planning System (Siemens, Erlangen, Germany) or RayStation version 8A with Monte Carlo planning (RaySearch Laboratories, Stockholm, Sweden) using the PBS algorithm. Plans were optimized separately at each center. The respective planning system and dose-volume histogram data were pooled for estimation of secondary malignancy risks and statistical analysis. Dose and fractionation schedule were determined based on Masaoka stage and resection status. Optimization was performed using constraints as proposed by the current guideline [10] while respecting the ALARA (as low as reasonably achievable) principle for at-risk OARs. 

### 2.2. Risk Estimation for Radiation-Induced Secondary Cancers

The risk of radiation-induced secondary cancer was estimated using two distinct, well established, radiobiological models, initially described by Dasu et al. [15] and Schneider et al. [16].

The Dasu model is also known as the “competition model”, as it describes the competition between the induction of carcinogenic mutations and cellular survival and further considers both treatment dose fractionation as well as non-uniformity of the dose distribution across the irradiated organ [15]. The Schneider model is based on the calculation of the organ equivalent dose (OED) [16]. The OED concept postulates that any two dose distributions in an organ are equivalent if they result in the same radiation-induced SM incidence. In the Schneider model, besides the induction of carcinogenic mutations and cellular survival, repopulation and repair are also taken into account to calculate the risk for inducing SM [16,17]. 

Data extracted from the dose–volume histograms (DVHs) for each treatment modality were used for risk calculation of radiation-induced SM.

#### Dasu Model

In short, the Dasu model is a linear-quadratic (LQ)-based model (Equation (1)).
(1)$${\mathrm{Total}\;\mathrm{risk}}_{\mathrm{organ}}=\frac{1}{\sum_{i}v_{i}}\underset{i}{\sum }v_{i}\times \left\{\left(\alpha_{1}D_{i}+\frac{\beta_{1}D_{i}^{2}}{n}\right)\times exp\left[-\left(\alpha_{2}D_{i}+\frac{\beta_{2}D_{i}^{2}}{n}\right)\right]\right\}$$
where *v_i_* is the volume of tissue exposed to dose *D_i_* applied in *n* fractions. The parameters *α*_1_ and *α*_2_ are illustrated in Table 1. For the Dasu model, the term “total risk” comprises the risk for development of any cancer, while the term “fatal risk” only includes secondary malignancies resulting in death. An *α*/*β* ratio of 3 was used for OARs. 

### 2.3. Schneider Model

The risk of inducing SM was also estimated utilizing the Schneider model, which is based on determination of OED (Equation (2) of [16]). The OED concept postulates that any two dose distributions in an organ are equivalent if they result in the same radiation-induced SM incidence.
(2)$$\mathrm{OED}=\frac{1}{N}\overset{N}{\underset{i=1}{\sum }}D_{i}e^{-\alpha_{org}\;D_{i}}$$
where *v_i_* and *D_i_* are defined as in the Dasu model above and the sum is taken over *N* dose calculation points, which represent the same constant volume of the organ.

Based on the OED, the incidence of a secondary malignancy $I^{org}$ was calculated using the equation Schneider suggested [18,19]. Here, $I_{0}^{org}$ is the organ specific cancer incidence rate and $\alpha_{org}$ is the specific sterilization parameter. Values used for this parameter are shown in Table 2.
(3)$$I^{org}\;=\;I_{0}^{org}\;\mathrm{OED}\;e^{-\alpha_{org}\;\mathrm{OED}}$$

### 2.4. Statistical Analysis

All statistical analyses were performed using the software SPSS 24.0 (IBM Corporation, Armonk, NY, USA), utilizing the non-parametric Wilcoxon signed-rank test for pairwise comparison of the groups. Significance was noted for two-tailed *p*-values of ≤0.05.

## 3. Results 

### 3.1. Patient and Treatment Characteristics

A total of seventeen patients with thymic malignancies treated at the two different centers were included in the analysis. The gender distribution was evenly spread between eight male and nine female patients with a median age of 58 years (range: 17–78 years). Patients were evaluated utilizing the Masaoka system with stages ranging from I to IVB. The median radiation dose was 54 Gy (RBE) with a range of 45 to 66 Gy (RBE) over a median of 27 fractions (range: 25–33 fractions). The median tumor size was 49 mm (range: 13–139 mm) with a median PTV of 394 cc (range: 154–1213 cc). Complete patient-, treatment-, and disease-specific characteristics are shown in Table 3. Comparison 3DCRT, IMRT and PBS plans were then generated utilizing the initial prescription dose and were optimized using the aforementioned planning software. Dosimetric information for a given patient was then extracted and compared between the three radiotherapy techniques. Thoracic organs at risk were then independently evaluated using the Dasu and Schneider models to estimate the difference in SM risk for each radiotherapy technique.

### 3.2. Risk Estimation of Secondary Malignancies: Dasu Model

Total and fatal SM risk were calculated using the Dasu model. There was a significant reduction in lung total SM risk using PBS–PBT relative to both 3DCRT (0.84% vs. 1.95%, *p* < 0.001) and IMRT (0.84% vs. 2.13%, *p* < 0.001); however, no significant reduction was observed between 3DCRT and IMRT (1.95 vs. 2.13, *p* = 0.055). Similarly, there was a significant reduction in esophagus total SM risk using PBS–PBT relative to both 3DCRT (0.57% vs. 0.96%, *p* = 0.001) and IMRT (0.57% vs. 0.96%, *p* < 0.001); however, no significant reduction was observed between 3DCRT and IMRT (0.96 vs. 0.96, *p* = 0.868). Total SM risk was independently calculated for each thoracic organ at risk using the three radiotherapy models is listed in Table 4.

For both the left and right breast, there was a profound reduction (2–3x lower) in total SM risk using PBS–PBT relative to both 3DCRT and IMRT. The bilateral breast SM risk was also significantly reduced when using IMRT versus 3DCRT. Finally, there was no significant difference observed in total thyroid SM risk between the three radiation modalities. A very similar pattern for fatal mortality risk induction using the Dasu model was observed and is illustrated in Table 4. Figure 1 depicts the calculated risks for the relevant thoracic organs for each of the seventeen patients using each radiation technique. 

### 3.3. Risk Estimation of Secondary Malignancies: Schneider Model

SM risk was also estimated using the Schneider model to calculate the organ-specific cancer incidence for each radiotherapy technique. Organ-specific SM risks were estimated per 10,000 patients per year per Gy using the Schneider model with patient-specific rates illustrated in Figure 2 and cohort averaged rates in Table 5. The results utilizing the Schneider model parallel those seen with the Dasu model. Proton therapy significantly reduced the risk of radiation-induced lung, breast, and esophagus cancer compared to both 3DCRT and IMRT (Table 5).

The calculated cancer incidence rate (per 10,000 patients per year per Gy) for the bilateral lungs was significantly lower for PBS at 1.49 relative to 2.88 for IMRT and 2.74 for 3DCRT (*p* < 0.001). Again, no significant difference was observed when comparing IMRT to 3DCRT for lung cancer risk. For the left and the right breast, the cancer incidence rate was significantly decreased with the use of PBS-PBT (0.81 and 0.55, respectively) compared to 3DCRT (2.15 and 2.26, respectively) and IMRT (1.68 and 1.72, respectively, *p* ≤ 0.001). For the esophagus, a significant reduction to 1.04 for PBS was observed compared to 1.56 for 3DCRT and 1.54 for IMRT (*p* ≤ 0.001). The use of IMRT compared to 3DCRT significantly reduced the cancer incidence rate for bilateral breast cancer (*p* = 0.035 and 0.044), but there was no significant reduction for the remaining OARs. Finally, similar to the Dasu model findings, no significant differences in cancer incidence rates for the thyroid gland were detected for PBS, 3DCRT, or IMRT.

## 4. Discussion

SM risk is a relatively uncommon long-term side effect of therapeutic radiation. Despite its rarity, it is oftentimes the side effect patients dread the most. The risk of SM is known to be elevated in the younger patient population as well as those with prolonged cancer survivorship such as those diagnosed with thymic malignancies. Moreover, it is well established that as a stochastic effect, SM can be induced even at low doses of radiotherapy, thus the principle of ALARA holds particularly true for this potentially deadly long-term toxicity [20]. As radiotherapy techniques have advanced, the radiation oncologist’s ability to conformally treat targets and evade normal structures has concomitantly evolved. Nevertheless, although advances such as IMRT have allowed for improvements in conformality, in many cases it has come at the expense of larger integral doses delivered to normal tissues in the low and intermediate dose range [21]. As such, the mitigation of SM risk still requires optimization. 

Proton therapy’s exploitation of the Bragg peak allows for a physical dose superiority and a remarkable reduction in integral dose exposure (~50%) relative to photon-based treatments [22,23]. Such reductions in integral dose delivery in the treatment of thymic malignancies may widen the therapeutic window in a disease site where, historically, radiotherapy benefit has been challenging to definitively establish in the stage II adjuvant setting [6]. By limiting normal tissue dose exposure down to zero, in many cases, secondary malignancies can be analogously reduced, at least theoretically. Nevertheless, clinical data supporting such a hypothesis are very difficult to accumulate given the rarity of the toxicity and the extended follow up required to obtain such data. 

In the present study, we offer advanced radiobiological estimations of SM risk using the Dasu and Schneider models with a pairwise comparison of 3DCRT, IMRT, and PBS. Our results demonstrate consistent reductions in SM risks with the use of PBS in comparison to both comparison X-ray-based techniques. These improvements with PBS were observed in SM risks of the lung, breast, and esophagus. Moreover, they were reproducible in both the Dasu and Schneider models with reductions in total, fatal, and predicted secondary malignancy rates per 10,000 patients per year per Gy. Ultimately, proton therapy halved the secondary malignancy risk for all thoracic OARs analyzed with the exception of the thyroid gland. This lack of improvement in thyroid SM risk is likely explained by the anatomical location of the organ relative to the anterior beam geometry used in the present study (Figure 3).

The aggregate risk reduction was most evident for breast tissue with an approximate risk of 16–18% for 3DCRT,11–12% for IMRT, and 3–5% with PBS-PBT according to the Dasu model. Nevertheless, the individual risk of the patients may differ substantially based on intrinsic anatomy of the patient as is depicted in Figure 3A–C. Given the significant reduction in SM risk of bilateral breast tissue, a strong consideration for proton therapy in young female patients requiring thoracic radiotherapy for thymic malignancies should be considered.

The literature regarding SM risk following treatment for thymic tumors is not surprisingly limited. Population-based cancer registry data published during an antiquated radiation therapy era demonstrated an increased SM risk for patients receiving radiotherapy for thymic malignancies. Interestingly, the secondary cancers observed were primarily in organs likely to be within the radiotherapy field (e.g., lungs and esophagus) [4,24]. Nevertheless, drawing treatment-related conclusions from population-based analyses is tenuous at best, particularly given the purported background increased risk of SM in the general thymoma patient population and the conflicting data with respect to radiation therapy [2,3,25].

The only other similar publication identified in the literature comes from Vogel et al. [19]. Here the authors explored the use of older proton technology (i.e., double-scattered proton beam radiotherapy) in ten patients diagnosed with thymoma. Comparative IMRT, sans comparative 3DCRT plans, were developed to calculate SM risk, though only the Schneider model was used in this study. The authors observed significant reductions in the risk of secondary malignancies across nearly all organs analyzed (i.e., lung, breast, esophagus, skin, and stomach) with the use of proton therapy relative to IMRT. The improvements observed by Vogel et al. are consistent with those observed in the present study using the Schneider and Dasu model. Moreover, the use of active scanning proton therapy in our analysis, relative to double-scattered PBT as in Vogel et al. with notably mitigated secondary neutron exposure could result in additional reductions in SM risks and also reflects the modern technology used at today’s proton centers. 

Limitations of the present study include its small patient numbers and retrospective analysis. Moreover, much of the improvement associated with PBS is dictated by intrinsic patient anatomy and the geometry of the proton beam(s) utilized. Variations in anatomy, both target volumes and organs at risk, can obviously significantly change dosimetric parameters between radiation modalities, thus influencing predicted SM risk. As such, careful evaluation by a treating radiation oncologist with proton experience is required to determine individual patient benefit. 

Furthermore, patients were treated at two different institutions by different physicians, which limits the comparability of data. While at Georgetown University, Washington, adjuvant treatment for thymic malignancies was standard with proton therapy, at Heidelberg University, 10 consecutive patients with thymic malignancies treated with adjuvant radiotherapy between December 2013 and September 2016 were included into the study. Thus, selection bias likely contributed less, at least once the patient had presented to the clinic. Additionally, statistical comparison was performed on a patient-by-patient basis to further minimize biases potentially caused by different institutions. Prior dosimetric studies have utilized very similar patient populations for retrospective analysis. Fundamentally, these patients present the characteristic anatomy and geometry of the disease population that is then utilized to calculate secondary malignancy risk using these novel radio biological parameters. 

Nevertheless, this is the largest publication of its kind and uses two novel radiobiological models to answer this important clinical question. Given the consistent reductions in secondary malignancy risks predicted by the Dasu and Schneider models for lung, breast, and esophagus organs, we advocate for the consideration of proton beam therapy in the treatment of patients with thymic malignancies, particularly those diagnosed in young female patients.

## 5. Conclusions

Proton therapy for patients diagnosed with thymic malignancies can yield dramatic dose reductions to adjacent thoracic OARs. We use radiobiological estimations of secondary malignancy risk using established Dasu and Schneider models and demonstrate that proton therapies halve the risk of secondary malignancies of the lung, breast, and esophagus relative to X-ray-based techniques including IMRT. As such, we advocate for the consideration of proton beam therapy in the treatment of patients with thymic malignancies, particularly those diagnosed in young female patients.

## Figures and Tables

**Figure 1 cancers-14-02409-f001:**
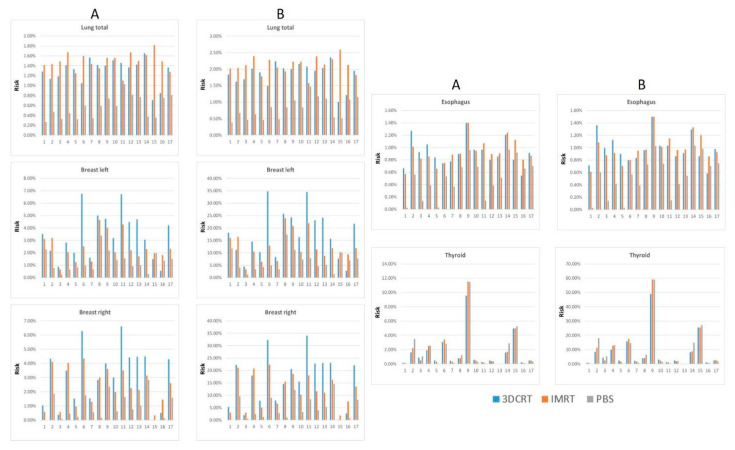
Calculated risk of total (**A**) and fatal (**B**) cancer induction for 3DCRT (blue) IMRT (red) and PBS (grey) plans for the respective organs at risk (lung, left and right breast, as well as esophagus and thyroid) according to the Dasu model for each patient.

**Figure 2 cancers-14-02409-f002:**
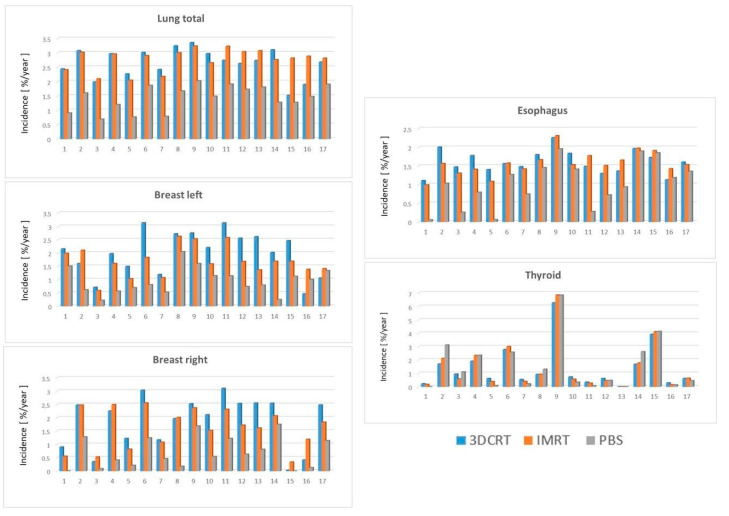
The organ-specific cancer incidence rates per 10,000 patients per year per Gy (lung, left and right breast, as well as esophagus and thyroid) according to the Schneider-model for 3DCRT (blue), IMRT (red), and PBS (grey) plans for each patient.

**Figure 3 cancers-14-02409-f003:**
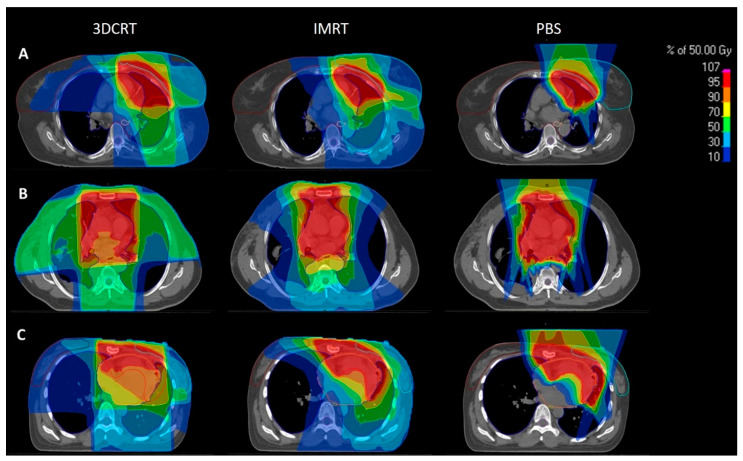
Three representative patients (**A**–**C**) with their comparative 3DCRT (first column), IMRT (second column), and PBS (third column) plans with color wash dose distribution. Compared to both the 3DCRT and IMRT plans, PBS plans enable dramatic reductions in integral dose exposure to adjacent OARs. 3DCRT: X-ray based 3D conformal radiotherapy, IMRT: X-ray based intensity-modulated radiotherapy, PBS: and active pencil beam scanned proton therapy.

**Table 1 cancers-14-02409-t001:** Risk coefficients (*α*_1_, second and third column) and the linear quadratic model parameter (last column) used for risk assessment for the different organs at risk.

Organ	*α*_1_ (Gy^−1^) Fatal Risk	*α*_1_ (Gy^−1^) Total Risk
**Lung**	0.0101	0.0144
**Breast**	0.0028	0.0144
**Esophagus**	0.0014	0.0015
**Thyroid**	0.0028	0.0144

The risk coefficients were taken from ICRP 103 according to Mondlane et al. [17].

**Table 2 cancers-14-02409-t002:** Organ specific incidence $I_{0}^{org}$ and sterilization parameter $\alpha_{org}$ for SM.

Organ	$I_{0}^{org}$ (per 10,000 Patients/year/Gy)	$\alpha_{org}$ (Gy^−1^)
**Lung**	1.68	0.129
**Breast**	0.78	0.08
**Esophagus**	0.61	0.274
**Thyroid**	0.75	0.033

The coefficients are based on Schneider et al. [18].

**Table 3 cancers-14-02409-t003:** Clinical patient characteristics.

Patient No.	Sex	Age	Masaoka Stage	WHO Type	R-Status	Max. Tumor Size [mm]	RT Total Dose [Gy]	Fractions
1	F	77	IIa	B1/B2	R0	40	50	25
2	F	23	I	B2	R0	13	50	25
3	F	58	IIa	B2	R0	37	50	25
4	M	42	IIa	AB	R1	37	54	27
5	M	71	IIa	B3	R0	30	50	25
6	M	56	III	AB	R0	80	50	25
7	F	53	IIa	B2	R0	17	50	25
8	F	44	IIa	B2/B3	R0	42	54	27
9	M	47	IVa	B3	R2	53	66	33
10	M	62	IVa	B1/B2	R2	99	66	33
11	F	69	I	B3	R0	65	54	30
12	M	70	II	B2	R1	59	54	30
13	F	78	IIA	C	R0	44	54	30
14	M	65	IVB	N/A	N/A	49	45	25
15	F	73	IVB	B2	R0	139	54	30
16	F	31	III	B2	R1	90	54	30
17	M	17	I	B2	R1	110	54	30

F = female, M = male, RT = radiotherapy, R = resection.

**Table 4 cancers-14-02409-t004:** Median fatal and total secondary risk values of the Dasu model in percentage for the different techniques.

Dasu Total	3DCRT (%)	IMRT (%)	PBS (%)	PBS vs. 3DCRT	PBS vs. IMRT	IMRT vs. 3DCRT
**Lung total**	1.95 (1.01–2.36)	2.13 (1.57–2.59)	0.84 (0.38–1.47)	*p* < 0.001	*p* < 0.001	*p* = 0.055
**Breast left**	16.38 (2.83–34.73)	11.32 (3.35–23.88)	5.18 (1.27–17.37)	*p* = 0.001	*p* < 0.001	*p* = 0.019
**Breast right**	17.94 (0.15–34.01)	11.59 (1.79–2.38)	3.21 (0–14.56)	*p* < 0.001	*p* < 0.001	*p* = 0.019
**Esophagus**	0.96 (0.58–1.5)	0.96 (0.61–1.5)	0.57 (0.03–1.03)	*p* = 0.001	*p* < 0.001	*p* = 0.868
**Thyroid**	3.06 (0–49.15)	2.51 (0–59.18)	2.03 (0–58.96)	*p* = 0.379	*p* = 0.796	*p* = 0.918
**Dasu Fatal**	**3DCRT (%)**	**IMRT (%)**	**PBS (%)**	**PBS vs. 3DCRT**	**PBS vs. IMRT**	**IMRT vs. 3DCRT**
**Lung fatal**	1.37 (0.71–1.65)	1.49 (1.1–1.82)	0.59 (0.27–1.03)	*p* < 0.001	*p* < 0.001	*p* = 0.055
**Breast left**	3.19 (0.55–6.75)	2.20 (0.65–4.64)	1.01 (0.25–3.38)	*p* = 0.001	*p* < 0.001	*p* = 0.019
**Breast right**	3.49 (0.03–6.61)	2.25 (0.35–4.35)	0.62 (0–2.83)	*p* < 0.001	*p* < 0.001	*p* = 0.019
**Esophagus**	0.89 (0.55-1.40)	0.90 (0.57-1.4)	0.53 (0.03–0.96)	*p* = 0.001	*p* < 0.001	*p* = 0.868
**Thyroid**	0.59 (0–9.56)	0.49 (0–11.51)	0.4 (0–11.46)	*p* = 0.379	*p* = 0.796	*p* = 0.918

**Table 5 cancers-14-02409-t005:** Median values (range) of the predicted secondary malignancy rates per 10,000 patients per year per Gy using the Schneider model for the different techniques.

Cancer Incidence Rates	3DCRT	IMRT	PBS	PBS vs. 3DCRT	PBS vs. IMRT	IMRT vs. 3DCRT
**Lung total**	2.74 (1.52–3.36)	2.88 (2.05–3.24)	1.49 (0.69–2.03)	*p* < 0.001	*p* < 0.001	*p* = 0.619
**Breast left**	2.15 (0.47–3.13)	1.68 (0.59–2.62)	0.81 (0.22–2.05)	*p* = 0.001	*p* < 0.001	*p* = 0.035
**Breast right**	2.26 (0.03–3.11)	1.72 (0.34–2.26)	0.55 (0–1.75)	*p* < 0.001	*p* < 0.001	*p* = 0.044
**Esophagus**	1.56 (1.11–2.26)	1.54 (0.99–2.23)	1.04 (0.05–1.96)	*p* = 0.001	*p* < 0.001	*p* = 0.463
**Thyroid**	0.83 (0.22–6.27)	0.61 (0.15–6.86)	0.79 (0–6.85)	*p* = 0.679	*p* = 0.959	*p* = 0.717

## Data Availability

The datasets generated for this study will not be made publicly available since national legislation and the terms of study ethics approval do not allow dataset sharing outside of the institutions participating in the analysis.

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
