# Peer review of "Secondary Malignancy Risk Following Proton vs. X-ray Radiotherapy of Thymic Epithelial Tumors: A Comparative Modeling Study of Thoracic Organ-Specific Cancer Risk"

_cancers, 2022, doi:10.3390/cancers14102409_

Round 1

Reviewer 1 Report

Overall:

This study is an application of two teoretical models that have already been teoretically applied in a comparable form for a variety of histologies. The question is in principle very interesting for the proton community and those of us interested. Here the authors report an etinity that is in principle not yet investigated with this model, but I consider the orginality of this work still limited. If the data presented were to be truly novel, it would be if it were combined with clinical experience. As such, the paper is merely a dosimetric comparison that shows exactly what one would have intuitively suspected, namely that normal tissue is dosimetrically better spared with proton therapy. After so many years of routine clinical use of proton therapy, however, I believe that clinical studies are now necessary to show whether this dosimetric advantage is also reflected in a clinical benefit. I also development of a second carcinoma to be much more complex than it is captured in the calculation examples. The authors already mention in their text that different histologies have an increased risk for second tumors, which is not captured in the calculation example.

Secondly, it could be shown in clinical experience that the risk of infield secondary malignancies could also be reduced by proton therapy. Again, since the study just does not go beyond a dosimetric comparison, this cannot be shown.

In detail:Please revise the outline of the publication. It seems irritating that the material, methods part, comes after the result part.

Methods:

Were all OAR contoured by the same physician?

Were all plans performed according to the same standards?

Was no CTV margin applied to patients after GTR? If so, isn't the target volume after GTR (based on imaging BEFORE op) larger than in patients with a small remnant?

Or is it correct that in patients with a small remnant, the preoperative contact surfaces were not irradiated.

Was no 4DCT applied. If this is because the data is historical, one has to wonder how the results are applicable to today's cohorts, since they would likely be planned with 4DCT and the ZV would expand to include an ITV.

How were the density uncertainties accounted for in the proton plan? Now it looks like the PTVs were both the same size. Yet for the density uncertainties, the ZV would have to be larger in the proton plan?

What did the authors do as motion management in the proton plans?

How was neutron scattering captured in the model?

In summary: I think the paper falls short of the possibilities of the question. From my point of view, a purely dosimetric study cannot adequately answer the important question.

Author Response

Please see answers in our attached and detailed cover letter with point by point answers. Thank you.

Reviewer 2 Report

Proton therapy is an advanced radiation technique but at the moment data supporting it are small and in the most cases retrospectives. This paper, despite the dosimetric nature of the study design, shows a possible advantage of proton therapy respect conventional radiotherapy in a specific subgroup of patients (female young patients). I believe that this paper should be taken into consideration for a pubblication given the current interest to define a role of proton therapy in mediastinal lesions.

In the paper submitted there is an inversion of paragraphs: material and methods are after results. 

Author Response

(The authors gave the same response as above.)

Reviewer 3 Report

This study uses existing mathematical models for radiation-induced secondary malignancies to compare IMRT, 3D-CRT, and PBS-PBT for adjuvant radiation therapy in the treatment of thymoma. 

The methods and results are clearly presented and the conclusions are supported by results from this study. This manuscript is appropriate for publication. 

Author Response

Thanks

Reviewer 4 Report

Thank you for the opportunity to review the manuscript.

This is an article that presents the calculation of the risk
of radiation-induced cancers after photon-based 3D conformal RT, intensity-modulated RT, and active pencil beam scanned proton therapy (PBS) in patients with thymic malignancies.

The paper adds some value to the current knowledge, has potential clinical applications, and discusses an important problem. 

However, I recommend introducing major changes in the manuscript before consideration for publication.

INTRODUCTION:
I recommend you modify the introduction to make it more concise. Currently, it is very chaotic, with some repetitions and unnecessary discussions. For example, you mention several times that thymoma is a rare disease, you write about thymoma, then jump into protons, and again into thymoma.
Please add data on recommendations for RT in thymoma and thymic carcinoma (American NCCN, European ESMO, etc.).
You stated that "One possible reason for the lack of overall survival improvement with radiotherapy may be the toxicity profile caused by utilization of more antiquated radiation techniques such as 3DCR" - could you give some examples from the literature, any references? Such a hypothesis requires a really long follow-up in analyzed cohorts. 

METHODS AND MATERIALS:
Was the treatment planning performed independently by at least two (blinded) medical physicists? How did you eliminate or reduce severe biases that may occur in such a study (i.e. selection bias, omission bias, measurement bias, etc.)? In my opinion, the risk of bias when you calculate treatment plans for retrospective patients is significant.
Did you calculate only step-and-shoot IMRT? Or also dynamic IMRT, especially VMAT? VMAT seems to be preferred in the case of thoracic tumors.

DISCUSSION:
Please discuss more extensively the limitations of the study (as above).

CONCLUSIONS:
Conclusions should be rewritten - you should only interpret finding from your study, not general knowledge. Did you find that "Thymic malignancies are rare tumors often diagnosed in healthy patients with long expected cancer survivorship"? No. Again "The role of radiotherapy particularly in the adjuvant setting for this disease site has historically been nebulous in part due to the longterm risks of radiation exposure to critical thoracic organs." - it is also not your finding. 

Author Response

Reviewer 4:

Introduction

We have updated the introduction to reflect current recommendations for radiotherapy in the context of thymoma and thymic carcinoma and restructured it according to the recommendations. Furthermore, we have added the requested reference (no. 8) for the statement: “One possible reason for the lack of overall survival improvement with radiotherapy may be the toxicity profile caused by utilization of more antiquated radiation techniques such as 3DCRT”.

Material and Methods

Treatment planning was performed independently by two parties.  These were independent of the initial proton therapy planners to avoid bias as a relates to creation of the x-ray based plan.

Regarding the possible biases and limitations we added a section to the discussion as stated below.

The reviewer is right, there are different types of IMRT which are different. Our plans were all calculated as VMAT. We added this to the M&M section

Discussion

Certainly, retrospective reviews of data have significant limitations as a relates to bias.  At Georgetown University, adjuvant treatment for thymic malignancies was standard with proton beam therapy.  At Heidelberg University, 10 consecutive patients with thymic malignancies treated with adjuvant photon radiotherapy between December 2013 and September 2016 were included into the study. Thus selection bias likely contributed less, at least once the patient had presented to clinic.  Additionally, statistical comparison was performed on a patient-by-patient basis to further minimize biases potentially caused by different institutions. That being said, prior dosimetric studies have utilized very similar patient populations for retrospective analysis.  Fundamentally, these patients present the characteristic anatomy and geometry of the disease population that is then utilized to calculate secondary malignancy risk using these novel radio biological parameters.

We have updated the limitations of the present study within the discussion section.

Conclusion

We have adjusted the conclusions based on the reviewer's recommendations.

Reviewer 5 Report

In  the Introduction  elaborate  on proton beam effect  on  human  tissue, Bragg effect and  distribution  of  energy given by protons vs. x-ray radiation

Elaborate more on mechanism proton  induced apoptosis of cancer  cells with  comparison to x-ray induced apoptosis related  to secondary fibrosis  of  tissue

Explain properties of  Proton beam radiation therapy in human tissue  and compare  with  Muon induced disruption of hydrogen bond networks in the AKT  gene activation in  cancer and why proton or muon have little dose exit in surrounding tissue

Discuss effectiveness and risk for secondary malignancy of x-ray, proton  or  muon  therapy

Equation of Dasu model  needs to be explained  in  details  especially how exponent   factor, influencing  the equation, also,  same for organ-equivalent dose  equation

Figure 3 Legend needs detailed explanation   

Author Response

Reviewer 5:

We have described proton therapy and the Bragg peak in more detail within the introduction.

The radiobiology of apoptosis as it relates to secondary fibrosis of the tissue is beyond the scope of the current manuscript and not relevant to the present study. This seems to be a very certain and complex question of radiobiological analysis and was never the topic of this clinical analysis and manuscript. This seems to be the primary research focus of the reviewer and is not appropriate for this analysis.

The consequences of Muon induced a disruption of hydrogen bond that works in the AKT gene activation is also beyond the scope of this manuscript and not relevant in the present study.

The description of the previous radiobiological model is complex and is now described more in detail in the Materials and Methods sections. The exponent factors are shown in the tables and referenced to the initial publications.

Figure 3 caption and the figure itself were extended and adapted to better highlight the statement of the figure.

Round 2

Reviewer 1 Report

The central question of the paper is the second tumor development by protons compared to photons. This question has been published countless times. Just because the authors now extend the same model to another disease does not make the data in any way superior. If thymus tumors have a specific intristic contribution to the development of second tumors, then this is not studied at all. Badve et al, could show that risk for second tumors is particularly high in thymic carcinomas. So there seems to be a particularly high intrinsic potential for the development of second tumors. There is no theoretical model that the authors use to integrate this in any way. So it doesn't really matter whether it's thymic carcinoma or any other malignancy in this region. I consider this to be a substantial weakness regarding quality of content. 
There are also no new methodological approaches that really distinguish the paper from all the countless published papers. 
Furthermore, I consider it a substantial weakness in a planning study that the patients were contoured and planned at different institutes by different people. It cannot be excluded that the shown differences are not the result of differences between the individual institutes. The fact that the authors didn't bother to recontour and plan the plans shows that, in my opinion, they didn't try carefully enough to minimize the risk of bias. 
I think that a pure planning study without a new theoretical approach without any clinical data, as it has already been shown infinitely many times, is too weak and also too irrelevant to stimulate an interest or discussion in the community. 
Nevertheless, I would like to thank the authors for their kind words and the constructive cooperation.  

Author Response

Reviewer #1

This reviewer references countless previous studies but does not provide any support for that statement.  To our knowledge, the literature regarding the risk for secondary malignancies in patients with thymic cancer is scarce. Therefore, we kindly request these "countless" studies be provided as supporting documentation. There is effectively one vaguely similar published study that is specific to thymic cancer and secondary malignancy risk that has been published in the past using older radiation techniques - not the novel techniques that we research here that are now more generalizability of the general proton facility in the modern era (e.g. active scanned proton therapy).  To presume that secondary malignancy risk is independent of disease site is not true.  Certainly, proton therapy is not uniformly superior from a secondary malignancy risk standpoint to x-ray based therapy for every disease site.  As such, it is critical to identify those disease sites in which that superiority can be shown.  This is obviously true in a lot of countries like the United States or Germany where insurance companies hesitate to provide coverage for proton therapy in the majority of disease sites.  The same insurance companies have made it very clear that additional research, such as the present study, is required to establish the efficacy of proton therapy relative to x-ray based therapy for given disease site. 

As such, this paper clearly highlights an important disease site that may benefit from the utilization of proton therapy. One of the fundamental purposes of utilizing proton therapy is the physical dose superiority, which is intrinsically related to geometry.  Moreover, the geometry of a certain disease site is, obviously, dramatically different.  Thus, as the reviewer suggests, extrapolation of prior publications in different disease sites is frankly not possible.

Why are we using a radiobiological model?

  • Thymic malignancy is extremely rare
  • Secondary malignancy risk is extremely rare
  • To clinically observe secondary malignancy risk requires decades of follow-up
  • To observe secondary malignancy risk with decades of follow-up in an extremely rare disease site is considerably challenging if not impossible
  • Novel radiobiological models are required to mitigate the aforementioned challenges and identify reasonable risk estimations in an extremely rare disease site, for an extremely rare complication, that would theoretically require decades of follow-up to identify

Why are we exploring thymic cancer as a disease site?

  • Thymic cancer occurs in patients who are typically healthy relative to other patients were diagnosed with thoracic malignancies
  • Healthier patients are likely to have prolonged survivorship relative to other cancer survivors
  • As survivorship increases, the cumulative risk of a secondary malignancy also increases
  • Proton therapy, given its physical dose superiority, may widen the therapeutic window relative to x-ray based therapy by reducing the risk of secondary malignancy
  • Thymic cancer also occurs predominantly in the anterior mediastinum, as such, the geometry may further widen the therapeutic window afforded to anteriorly weighted proton beams

Has this been published extensively?

  • No
  • There is only one other publication that is looked at this topic and it was utilizing an older form of proton radiation

Additional data should be published to support the use of PBT in this disease site. The reviewer is right, that as we used real patient data from two different institutions, this might have caused bias. However, this potential bias is minimized by applying the Wilcoxon signed rank test, which primarily compares the differences between a photon and a proton plan on an individual patient level and not as a sum for all patients. This fact was added to the limitation section in the manuscript.

Furthermore, we adapted the introduction to take all the mentioned points into consideration.

Reviewer 4 Report

The authors addressed all comments made.

Reviewer 5 Report

corrected as recommended

discussed as recommended

now  can  be  accepted  for  pulication